# Syphilis Infections, Reinfections and Serological Response in a Large Italian Sexually Transmitted Disease Centre: A Monocentric Retrospective Study

**DOI:** 10.3390/jcm11247499

**Published:** 2022-12-18

**Authors:** Valentina Marchese, Giorgio Tiecco, Samuele Storti, Melania Degli Antoni, Stefano Calza, Maurizio Gulletta, Francesca Viola, Emanuele Focà, Alberto Matteelli, Francesco Castelli, Eugenia Quiros-Roldan

**Affiliations:** 1Unit of Infectious and Tropical Diseases, Department of Clinical and Experimental Sciences, ASST Spedali Civili di Brescia and University of Brescia, 25123 Brescia, Italy; 2Unit of Infectious and Tropical Diseases, ASST Spedali Civili di Brescia, 25123 Brescia, Italy; 3Unit of Biostatistics and Bioinformatics, Department of Molecular and Translational Medicine, University of Brescia, 25123 Brescia, Italy

**Keywords:** syphilis, reinfections, HIV, serological response, serofast, serological non-responder

## Abstract

Background: Syphilis infection does not confer definitive and protective immunity against reinfection, and crucial aspects of repeated episodes of syphilis are far from being understood, especially among people living with HIV (PLWH). Methods: In order to explore the burden of syphilis in a large cohort of HIV-negative patients and PLWH, this retrospective study describes the demographics, clinical presentation and treatment outcome of patients with syphilis treated at our clinic from 2013 to 2021. Results: Within the study period, 1859 syphilis episodes (827, 44.5% first infections and 1032, 55.5% reinfections) were recorded. A total of 663 patients, of whom 347 (52%) had PLWH, were considered. Syphilis was mostly diagnosed in males (77%) and European (79%) patients. More than half of syphilis episodes were recorded during the late latent stage (64%) or during follow-up/screening visits for other diseases, while symptomatic stages led to a diagnosis in almost half of HIV-negative patients (*p* < 0.001). PLWH with syphilis infection were predominantly homo/bisexual (*p* < 0.001). A significantly higher rate of syphilis reinfection was observed in PLWH, who also demonstrated a higher range of subsequent episodes. The serofast state was found to be similar at the 6- and 12-month follow-up visits. The multivariate analysis carried out in the HIV-positive group showed that an RPR titre >1:16 was an independent predictor for serological non-response. Conclusions: Syphilis reinfections are predominantly diagnosed in HIV-positive MSM. The high rate of asymptomatic presentation among PLWH supports the role of periodical syphilis screening. In PLWH, the only baseline factor associated with an increased risk of non-response was an RPR titre >1:16, while assessment at 12 months after treatment increased the possibility of detecting a serological response, indicating that PLWH have a slower serological response to treatment.

## 1. Background

Syphilis is one of the oldest sexually transmitted diseases, for which curative and inexpensive treatment is available [1]. However, the annual rate of primary and secondary syphilis has risen in recent years, both in Europe and the USA [2]. In Italy, the estimated notification rate is 2.5/100,000, and syphilis disproportionately affects men, with an overall gender ratio of 7:1 (male-to-female) in 2018 [3]. Previous Italian national data (1991–2017) confirm the major burden in males, demonstrating that they account for 91% of all primary or secondary cases and 65% of latent syphilis [4].

During the last two years of the severe acute respiratory syndrome coronavirus 2 (SARS-CoV-2) pandemic, community containment and social distancing measures might have affected the circulation of STDs [5]. However, few and controversial data are available. A recent monocentric study carried out in Italy has shown that despite the lockdown and the fear of SARS-CoV-2 infection, risky sexual activity has not diminished; likewise, the risk of syphilis remains, raising concern for community health [6].

Syphilis is frequently encountered among people living with HIV (PLWH), with a non-uniquely defined prevalence; in Europe, the rate of HIV and syphilis coinfection (either known or newly diagnosed) is 24% overall and may reach 35% when specifically considering the bisexual population and other men who have sex with men (MSM) [2,7]. Moreover, syphilis facilitates both HIV transmission and HIV acquisition, and with the consolidation of the evidence that a stable virological suppression of plasmatic HIV-RNA does not lead to HIV transmission (U = U or undetectable = untransmittable), this bond is intended to last [1]. Coinfection is dangerous because HIV infection may modulate the clinical presentation and the serologic response to syphilis treatment [7]. PLWH with a syphilis infection seem to have a higher risk of developing neurosyphilis and mount an abnormal serological response [8,9]. Many diagnostic and therapeutic challenges remain, as there are currently no controlled trials focused on syphilis–HIV coinfection management. Standard HIV care should always include regular syphilis serology, but new models for testing and prevention will be crucial next steps in controlling this coinfection [10].

Syphilis infection does not lead to immunity against reinfection, making repeated episodes of syphilis a concrete reality, especially in MSM with a high rate of partner change [11]. Moreover, data regarding serological response in patients with syphilis reinfection are discordant, and crucial aspects of the interpretation of non-treponemal test trends are far from being resolved [1]. The non-uniquely defined serological response in syphilis reinfections, together with the altered immunologic response in PLWH, represents a challenge for clinicians.

In order to explore the burden of syphilis in a large cohort of HIV-negative patients and PLWH, this retrospective study describes the demographics, clinical presentation and treatment outcomes of patients with syphilis treated at our clinic from 2013 to 2021. It also aims at assessing possible factors associated with serological response to treatment for syphilis reinfections among PLWH, for whom a higher frequency of reinfection was recorded during the study period. We also aimed to assess relevant factors associated with a serological response to treatment and syphilis reinfection among our HIV/syphilis coinfected population.

## 2. Materials and Methods

### 2.1. Study Population and Setting

The study was conducted at the Department of Infectious Disease of the ASST Spedali Civili, Brescia, Italy. In 2021, our department provided care to 3841 PLWH and hosted a clinic for sexually transmitted diseases (STDs) as part of the Italian Sentinel Surveillance System.

We retrospectively evaluated all cases of syphilis reported by our clinic to the Italian Sentinel Syphilis Surveillance System from January 2013 to December 2021, matching them with our electronic health record system. Incomplete or uncertain records were excluded. The study population was split into 2 groups: the HIV-positive group and HIV-negative group or control group, according to the presence of HIV/syphilis coinfection. All cases of syphilis (first diagnosis or reinfection), either among PLWH or HIV-negative patients aged >18 years, were included.

### 2.2. Data Collection

Demographic and clinical/laboratory characteristics were collected from medical records. The variables included sex, age, area of origin according to the United Nations (UN) classification [12], sexual orientation, HIV infection and, among PLWH, HIV viral load and lymphocyte CD4+ T-cell count. For detected cases, data regarding clinical classification (primary, secondary, tertiary, early latent, late latent or neurosyphilis) and reasons for testing were collected (screening for PLWH, pregnancy, other STDs or partner tracing or presence of symptoms). Incomplete or uncertain cases were excluded. Moreover, only records with at least one follow-up consultation at 6 and/or 12 months after treatment, reporting signs or symptoms and non-treponemal test results were included to assess the serological response to treatment.

### 2.3. Clinical Management

We considered indirect methods to assess a syphilis diagnosis; TPHA (Treponema pallidum haemagglutination assay) as a treponemal test and RPR (rapid plasma reagin) as a non-treponemal test were the principally used assays. PLWH underwent syphilis screening at their first HIV clinical evaluation with both treponemal and non-treponemal tests. Patients with positive results received the appropriate supervised treatment. Screening was repeated annually (with an RPR test being used only in previously treated patients), or earlier in the case of reported symptoms or at-risk sexual intercourse. A thorough anamnesis and physical examination were performed in the case of a syphilis diagnosis to assess its stage, even performing a lumbar puncture according to current ongoing guidelines if needed [13]. All patients received appropriate treatment (mostly benzathine penicillin or doxycycline in the case of allergy). In HIV-negative individuals, indications for testing for syphilis infection in Italy include clinical suspicion, contact with syphilis, pregnant women and risk factors for sexually transmitted diseases. Follow-up visits were scheduled at 6 months and 12 months after treatment. During these visits, clinicians assessed risk factors for syphilis re-exposure and serological data were obtained to monitor the RPR titre trends. Patients with a serological non-response or a serofast status were asked for risk factors, and in doubtful cases, a new treatment was given.

### 2.4. Definitions

A first syphilis case was defined as a patient with positive treponemal and non-treponema tests without a previous history of treated syphilis infection whether in combination with clinical signs of syphilis or not. Reinfection was defined in patients having more than 2 diagnoses of syphilis infection during the study period in the case of having been previously diagnosed as having syphilis, having received adequate treatment and having demonstrated a seroreversion from negative to positive or a ≥4-fold increase in non-treponemal titre. Patients with undocumented treatment during the study period and patients with missing, incomplete or uncertain data were excluded from the study.

In accordance with the current literature, we considered patients with a complete seroreversion in RPR titres (from positive to negative) or with a ≥4-fold decline in nontreponemal antibody titres as serological responders [13,14]. Patients who, after an effective treatment, showed a ≤4-fold titre decline were defined as serological non-responders, whereas patients with a persistently reactive RPR titre despite adequate treatment and an initial ≥4-fold decline were defined as serofast.

### 2.5. Ethical Aspects

The study protocol received ethical approval from the Ethics Committee of the province of Brescia (code number NP 4847). All data were collected and analysed according to current Italian laws for the management of sensitive data and principles of the Declaration of Helsinki. The standard clinical practice at all healthcare services included providing thorough information and receiving verbal consent to any of the offered practices.

### 2.6. Statistical Analysis

Descriptive analysis of the characteristics of the two cohorts at baseline was performed. The median values and interquartile ranges (IQRs) were used to describe numerical variables, while the counts and percentages were employed for qualitative variables. A chi-square (χ2) test with *p*-values computed using Monte–Carlo simulations (B = 2000) or the Wilcoxon–Mann–Whitney test were used to compare the groups for categorical or continuous variables, respectively.

The number of reinfections was modelled using a generalised linear model (GLM) with a negative binomial family. The results are reported as incidence rate ratio (IRR) estimates and corresponding 95% confidence intervals (CIs 95%). Serological non-responses or serofast statuses were modelled as binary outcomes in a longitudinal setting using generalised linear mixed models (GLMMs) with a binomial family (also known as logistic regression). The results are reported as odds ratios (ORs) and corresponding 95% confidence intervals (CIs 95%). All tests were two-sided (apart from χ2), and a 5% significance level was assumed.

## 3. Results

### 3.1. Prevalence of Syphilis Episodes in the Study Period

In the study period, 1859 syphilis episodes were recorded. A total of 827 patients were diagnosed with a first episode of syphilis infection (44.5%) and divided according to their HIV infection status (Table 1). Between 2019 and 2021, we observed a decrease in first syphilis infection diagnoses among HIV-negative subjects. Approximately 1032 (55.5%) syphilis reinfections were recorded, especially in the HIV-positive population.

Approximately 164 (19.8%) records regarding first syphilis infections were incomplete. Thus, 95 (11%) HIV-negative and 69 (8%) HIV-positive patients were excluded due to missing data (Figure 1), whereas 650 (63%) subsequent episodes of syphilis infection were excluded due to patients being lost to follow-up. As shown in Figure 1, 61 HIV-negative and 209 HIV-positive patients had at least 1 new episode of syphilis during the follow-up visits at 6 and/or 12 months. A total of 54 and 328 episodes of syphilis reinfection were recorded among HIV-negative and HIV-positive patients, respectively.

### 3.2. Demographic and Clinical Characteristics of Included Patients

The study included 663 patients, 316 (48%) HIV-negative and 347 (52%) HIV-positive patients with a median CD4 count of 295.5 cells/mcL (range 4 to 1038). As shown in Table 2, syphilis was mostly diagnosed in males (508, 77%) with a median age of 39 years old (IQR, 12–83 years). Most patients were of European origin (522, 79%) and self-declared a homo/bisexual orientation (299, 45%). More than half of syphilis episodes were recorded during the late latent stage (424, 64%), followed by the secondary (107, 16%), primary (63, 10%) and early latent (53, 8%) stages. Neurosyphilis was observed in 16 (2%) episodes. Most syphilis episodes (426, 64%) were detected during follow-up/screening visits for other diseases (HIV, other STIs or partner tracing), whereas (237) 36% were detected due to the appearance of symptoms (skin rash, chancre or neurologic symptoms).

We found statistical differences between the HIV-positive group and the control group in several aspects of the demographic analysis. PLWH with syphilis infection were predominantly (260, 75%) homo/bisexual (*p* < 0.001), while most heterosexuals with syphilis (184, 58%) were in the control group. Almost half (163, 53%) of HIV-negative syphilis infections were diagnosed during symptomatic stages, while, considering the longer median follow-up (88.9 months, *p* < 0.001), PLWH were diagnosed mainly with syphilis (278, 80%) due to screening/follow-up reasons during asymptomatic stages (*p* < 0.001). Regarding syphilis reinfection, a significantly higher rate (*p* < 0.001) was observed in the HIV-positive group (209, 60%), who had a higher rate of subsequent episodes (1 to 7), whereas sporadic (61, 19%) episodes of syphilis reinfections were recorded in the control group.

Among PLWH, the logistic regression model to assess factors associated with the number of syphilis reinfections showed a protective association with age (IRR 0.65, 95% CI 0.47–0.91 for range 40–50 years, *p* = 0.011 and IRR 0.55, 95% CI 0.35–0.86, *p* = 0.009 for age > 50 years), while being homosexual/bisexual increased the risk of syphilis reinfections (IRR 1.52, 95% CI 1.09, 2.18, *p* = 0.017). In our model, gender, geographical origin and CD4 nadir were not associated with an increased risk of syphilis reinfection in PLWH (Table 3).

### 3.3. Serological Response to Treatment in Syphilis Reinfection

Among the patients with syphilis reinfection effectively treated according to their syphilis stage, no significant statistical difference was observed in the serological response between PLWH and HIV-negative patients at 6-month and 12-month follow-up visits (*p* = 0.2 at 6 months, *p* = 0.1 at 12 months). Among PLWH, the serologic response and seroreversion rate increased over time, while the serological non-response rate decreased during the follow-up assessment at 6 and 12 months after treatment. Conversely, the serofast status rate was found to be similar at both the 6- and at 12-month follow-up in PLWH, without any significant difference with the control group (Table 4).

Multivariate analysis was performed considering syphilis reinfection in the HIV-positive group. In our study, an RPR titre > 1:16 was an independent predictor (OR 2.26, 95% CI 1.14–4.51, *p* = 0.020) of serological non-response, regardless of sex, age, CD4 cell count at diagnosis or HIV viral load (Table 5). On the other hand, a follow-up evaluation at 12 months rather than earlier (6 months) statistically reduced the risk of serological non-response in PLWH (OR 0.38, 95% CI 0.22–0.65, *p* < 0.001). Among all the risk factors considered, none was statistically associated with the serofast state (Table 5).

## 4. Discussion

This was a large monocentric retrospective study including over 1800 episodes of syphilis infection describing a cohort of 663 people diagnosed with a first syphilis infection, dividing them into two groups according to their HIV status and focusing on serological response in syphilis reinfection among PLWH. As previously described, syphilis infection and homo/bisexual orientation were also often accompanied by HIV coinfection in our cohort [14]. Furthermore, the introduction of successful antiretroviral therapy and the general assumption among PLWH of U = U have induced an alarming decline in the adoption of safe sexual behaviours [10].

During the severe acute respiratory syndrome coronavirus 2 (SARS-CoV-2) pandemic, community containment and social distancing measures might have affected the circulation of STDs. A recent monocentric study showed that despite the lockdown and the fear of SARS-CoV-2 infection, risky sexual activity has not diminished, nor has unprotected sexual intercourse among the increasing number of PrEP users (pre-exposure prophylaxis). Thus, the occurrence of syphilis has continued unabated [6,15]. However, in a COVID-19-prioritised era, an inevitably downscaled number of individuals seeking medical attention might also partially justify possible incidence reductions in STD diagnoses [16]. In our centre, a statistical difference was found in the annual trend of syphilis reinfections among HIV-positive and HIV-negative patients. The minimal non-significant decrease in diagnoses of syphilis among PLWH observed during 2020 and 2021 is also in line with the maintained continuum of care in PLWH with few missed visits, but with a reduction in new HIV diagnoses observed in our centre during the SARS-CoV-2 pandemic [17,18]. Interestingly, national epidemiological data report a 20% reduction in latent syphilis diagnoses in 2020 compared to 2019, and a 5% reduction in primary and secondary syphilis [19]. Further studies are needed to clarify the link between lockdown and sexual habits and the risk for syphilis.

Syphilis infection does not lead to immunity against reinfection, and repeated episodes of syphilis occur predominantly in PLWH, especially in an asymptomatic form, highlighting the crucial role of periodical screening [20,21]. In our centre, PLWH are screened for syphilis at least annually, regardless of their symptoms or risk factors, allowing for an earlier diagnosis and intercepting repeated episodes. In our cohort, syphilis reinfections in PLWH ranged from one to seven episodes. It is known that MSM with HIV infection present high rates of syphilis coinfection and reinfection [22]. In particular, MSM report a syphilis reinfection rate higher than one in five men (71/323; 22%), while in PLWH, approximately 21.8% of coinfected patients experience a reinfection [23,24]. The higher percentage of syphilis among HIV-negative females with respect to HIV-positive females (44% vs. 4%) in our cohort reflects an occasional diagnosis of syphilis during the pregnancy screening programs.

A recent study with a large cohort showed that factors associated with more than one episode of syphilis were sex (male) (OR = 4.28), age (OR = 1.02), homosexual/bisexual orientation (OR = 2.29) and absence of STI symptoms at the time of syphilis diagnosis (OR = 1.70) [25]. The extent of sexually risky behaviour over time is the strongest risk factor for repeated syphilis episodes when compared to other indicators (antiretroviral regimen or immunological status) [26]. The identification of at-risk adults and adolescents is crucial to improve syphilis screening strategies, as recommended by the latest USPSTF (US Preventive Services Task Force) statements [27,28,29]. While the interpretation of the indirect methods for syphilis diagnosis is relatively straightforward in patients without a prior history of syphilis, it becomes more complex in the case of reinfections, as treponemal antibodies mostly remain positive for their entire life cycle, tests lack sensitivity in early infection (as well as when patients are contagious), and non-treponemal titres often lead to difficult-to-interpret serological responses after treatment [29].

Although there are no controlled clinical trials focused on optimising the treatment of syphilis, all our patients were treated following literature recommendations based on laboratory results, expert opinions, clinical cases and experience [30]. HIV infection did not influence the treatment regimen used and, in line with the most recent literature reviews, no additional antibiotic doses were used in the case of anomalous serologic responses [31,32].

The main finding of our study regards the serological response to treatment in syphilis reinfections; no significant statistical difference was observed in the serological response between PLWH and HIV-negative patients. Like other authors, we were unable to find an association between the CD4 count, HIV viral load and serological response [33,34,35]. Serological response to treatment in coinfection syphilis/HIV is controversial as unusual serologic responses might be recorded [36]. However, our results are consistent with the current literature, since the majority of PLWH infected with syphilis achieved an adequate serologic response [36]. PLWH may just take longer to reach an adequate serologic response after treatment [36]; in our study, seroreversion and serological non-response rates in PLWH raised and decreased, respectively, over time. HIV-related immunodeficiency was hypothesised as the cause of this controversial response [37]. In our study, an apparently overall better serological response was found in PLWH; the primary explanation for this is that PLWH are chronically evaluated in our centre with scheduled visits every 6 months. This increases the detection rate of syphilis infection at earlier stages, which is linked to a better serological response [38].

A decline in the serofast status rate over time was not confirmed in our analysis. However, there is a paucity of information about the serofast status in the literature. In a recent analysis, HIV coinfection was associated with a slower serological cure, and while just 37% seroreverted within a year, more than 60% still had a positive RPR after 1 year of follow-up [39]. Here, the serofast status rate was found to be similar at 6 and 12 months, but this does not mean that this trend must be confirmed in a more prolonged observational study (at a hypothetical 18-month or 24-month follow-up visit). In our clinical practice, patients with a serofast status or a serological non-response after treatment are asked about risk factors for new syphilis exposure and, in doubtful cases, a new treatment is given. It is univocally stated that neither the choice of therapy nor the posology (single dose vs. three doses of intramuscular benzathine benzylpenicillin) in PLWH has any influence on the serofast state in early syphilis [40].

The risk factors for serofast status in HIV coinfection are only partially understood, especially in the case of syphilis reinfections [40]. As previously described, in our analysis, the serofast state in subsequent episodes of syphilis infection was not statistically associated with gender, age, CD4 at diagnosis or HIV-RNA or RPR titre in PLWH [41]. Curiously, baseline RPR titres ≤ 1:16, CD4 counts < 350 cells/µL, untreated HIV infection and a previous syphilis history are considered predictors for persistent non-treponemal titres [42,43].

Lastly, the multivariate analysis carried out among syphilis reinfections in PLWH showed that both a high RPR titre (>1:16) and an earlier follow-up evaluation (6 months rather than 12 months) after appropriate treatment are independent risk factors for serological non-response. In PLWH, repeated episodes of syphilis are associated with higher non-treponemal titres, influencing the serological outcome after treatment, whereas an adequate titre response is generally obtained earlier in HIV-negative people [37]. In addition, patients with a lower baseline RPR titre during syphilis infection seem to require a longer period to achieve a serological response (252 days for RPR titre ≤1:8, 78 days for RPR titres from 1:16 to 1:32 and 53 days for RPR titres ≥1:64, respectively; *p* < 0.001). It is unlikely that an earlier evaluation (at 6 months after treatment) is associated with complete seroreversion [39]. Although RPR is routinely used to monitor for a syphilis reinfection, our results suggest a possible role of a high RPR titre (>1:16) as a predictor of serological non-response. The ab initio role of the RPR titre in syphilis management is controversial, and further studies are needed to confirm our finding.

The findings of our study should be seen in light of some limitations. First, it goes without saying that retrospective studies are subject to innate bias. The data were collected retrospectively from the patients’ electronic medical records, and in some cases, they were lacking. Second, the study was carried out in a single healthcare centre, and the conclusions drawn might not be representative of the general population. In our database, data for sexual risk behaviours were not always recorded, and a proportion of patients, especially in the HIV-negative group, were lost to follow-up, reducing the number of evaluable data on treatment response. Lastly, 24-month follow-up visits are routinely scheduled in our centre, but the available data were not sufficient to be considered in this study. Nonetheless, the study’s main strength is the large sample size and the comparability between the HIV-positive group and the control group. Moreover, a single health centre study ensures a unique interpretation of the most recent guidelines, resulting in a specific and shared way of treating patients and scheduling follow-up visits. Our findings should, therefore, be confirmed by further studies.

## 5. Conclusions

Syphilis rates have continued to rise in recent years. To control this epidemic, patients at risk must follow up with periodical screening, and tests must be correctly interpreted to provide appropriate treatment. Interpretations of syphilis serology can be challenging, and misinterpretation may result in undertreatment or overtreatment. HIV infection status should not be considered a factor influencing the choice of treatment or altering the interpretation of the serological results, even in the case of repeated episodes of syphilis. Regarding the serological response to treatment, PLWH probably need more time to obtain an adequate response, once more confirming the importance of chronic follow-up both to assess the response to treatment and to track reinfection earlier.

## Figures and Tables

**Figure 1 jcm-11-07499-f001:**
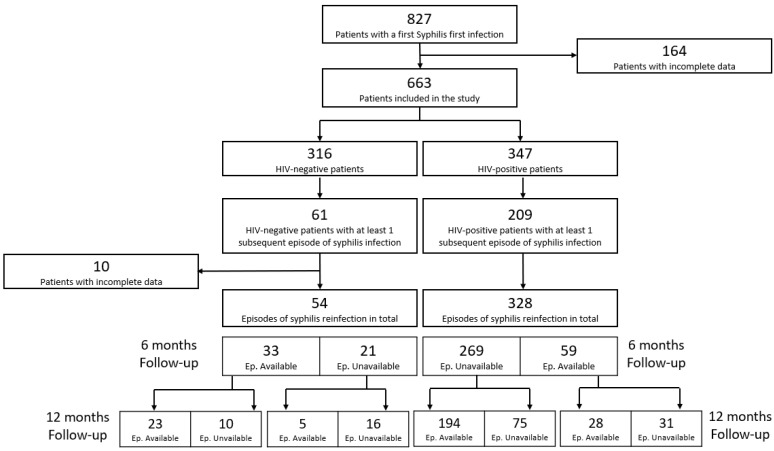
Study design (Ep. = episodes).

**Table 1 jcm-11-07499-t001:** Number of first syphilis infections and reinfections per year during the study period.

	First Syphilis Infections		Syphilis Reinfections	
Variables	HIV-Negative	HIV-Positive	HIV-Negative	HIV-Positive
**Year, n (%)**				
2013	51 (12%)	35 (8.4%)	14 (9.9%)	70 (7.9%)
2014	48 (12%)	43 (10%)	27 (19%)	107 (12%)
2015	48 (12%)	52 (12%)	30 (21%)	118 (13%)
2016	54 (13%)	68 (16%)	16 (11%)	123 (14%)
2017	54 (13%)	58 (14%)	18 (13%)	95 (11%)
2018	60 (15%)	40 (9.6%)	9 (6.3%)	89 (10%)
2019	38 (9.2%)	40 (9.6%)	13 (9.2%)	106 (12%)
2020	24 (5.8%)	38 (9.1%)	7 (4.9%)	115 (13%)
2021	34 (8.3%)	42 (10%)	8 (5.6%)	67 (7.5%)

**Table 2 jcm-11-07499-t002:** Demographic characteristics of the study population.

	HIV-Negative	HIV-Positive	Total	*p*-Value
**Number of patients with syphilis infection (n, %)**	316 (48%)	347 (52%)	663	
**Gender**				**<0.001**
Males (n, %)	176 (56%)	332 (96%)	508 (77%)	
Females (n, %)	140 (44%)	15 (4%)	155 (23%)	
**Age**				**0.026**
Median (Minimum, Maximum)	36 (12–83)	40 (12–75)	39 (12–83)	
**Geographical origin**				**<0.001**
Europe (n, %)	230 (73%)	292 (84%)	522 (79%)	
Other (n, %)	86 (27%)	55 (16%)	141 (21%)	
**Sexual orientation**				**<0.001**
Homosexual/Bisexual (n, %)	39 (12%)	260 (75%)	299 (45%)	
Heterosexual (n, %)	184 (58%)	73 (21%)	257 (39%)	
Prefer not to disclose (n, %)	93 (30%)	14 (4%)	107 (16%)	
**Syphilis stage during first infection**				**<0.001**
Late Latent (n, %)	228 (78%)	158 (51%)	386 (64%)	
Other (n, %)	63 (22%)	150 (49%)	213 (36%)	
Missing (n)	25	39	64	
**Reasons for syphilis test**				**<0.001**
Screening/Follow-up (n, %)	148 (47%)	278 (80%)	426 (64%)	
Symptom onset (n, %)	168 (53%)	69 (20%)	237 (36%)	
**Patients with syphilis reinfections**				**<0.001**
Yes (n, %)	61 (19%)	209 (60%)	270 (41%)	
No (n, %)	255 (81%)	138 (40%)	393 (59%)	
**Number of syphilis episodes per person**				**<0.001**
Median (Minimum, Maximum)	1 (1–3)	1 (1–7)	1 (1–7)	

**Table 3 jcm-11-07499-t003:** Generalised linear model to assess factors associated with the number of syphilis reinfections among PLWH.

Characteristic	IRR ^1^	95% CI ^1^	*p*-Value
**Gender**			
Female	—	—	
Male	2.44	0.98, 8.13	0.090
**Age (years)**			
≤30	—	—	
30–40	0.85	0.63, 1.15	0.3
40–50	**0.65**	**0.47, 0.91**	**0.011**
>50	**0.55**	**0.35, 0.86**	**0.009**
**Geographical origin**			
Other	—	—	
Europe	0.98	0.72, 1.34	0.9
**Sexual orientation**			
Heterosexual	—	—	
Homosexual/Bisexual	**1.52**	**1.09, 2.18**	**0.017**
**Nadir CD4**	0.98	0.91, 1.06	0.5

^1^ IRR = incidence rate ratio, CI = confidence interval.

**Table 4 jcm-11-07499-t004:** Serological response to treatment after syphilis reinfection at 6- and 12-month follow-up (SNR = serological non-response).

	HIV-Negative	HIV-Positive	*p*-Value
**Reinfection Episodes (n, %)**	54 (14%)	328 (86%)	
**Serological response at 6 months**			0.2
Yes (n, %)	13 (39%)	139 (52%)	
No or SNR (n, %)	20 (61%)	130 (48%)	
Missing (n)	21	59	
**Serological response at 12 months**			0.1
Yes (n, %)	14 (50%)	147 (66%)	
No or SNR (n, %)	14 (50%)	75 (34%)	
Missing (n)	26	106	
**Seroreversion rate at 6 months**			0.8
Yes (n, %)	6 (21%)	62 (23%)	
No (n, %)	26 (79%)	207 (77%)	
Missing (n)	21	59	
**Seroreversion rate at 12 months**			0.3
Yes (n, %)	7 (21%)	70 (32%)	
No (n, %)	22 (79%)	152 (68%)	
Missing (n)	26	106	
**Serofast status at 6 months**			0.2
Yes (n, %)	8 (24%)	94 (35%)	
No (n, %)	25 (76%)	175 (65%)	
Missing (n)	21	59	
**Serofast status at 12 months**			0.8
Yes (n, %)	10 (36%)	86 (39%)	
No (n, %)	18 (64%)	138 (61%)	
Missing (n)	26	106	

**Table 5 jcm-11-07499-t005:** Risk factors associated with a serological non-response and a serofast status in syphilis reinfection among people living with HIV.

	Serological Non-Response			Serofast Status		
Characteristic	OR	95% CI ^1^	*p*-Value	OR	95% CI ^1^	*p*-Value
**Time**						
6 months	—	—		—	—	
12 months	**0.38**	**0.22, 0.65**	**<0.001**	1.40	0.87, 2.25	0.2
**Sex**						
Female	—	—		—	—	
Male	2.67	0.25, 28.4	0.4	1.01	0.13, 8.06	>0.9
**Age**						
≤50 years	—	—		—	—	
>50 years	0.58	0.20, 1.67	0.3	1.17	0.47, 2.91	0.7
**Previous Episodes**	1.24	0.96, 1.60	0.11	0.92	0.72, 1.18	0.5
**CD4 at Diagnosis**						
>350 cells/mcL	—	—		—	—	
≤350 cells/mcL	1.03	0.35, 3.00	>0.9	1.24	0.47, 3.24	0.7
**HIV-RNA**						
Negative	—	—		—	—	
Positive	0.89	0.48, 1.63	0.7	0.97	0.56, 1.69	>0.9
**RPR titre**						
≤1:16	—	—		—	—	
>1:16	**2.26**	**1.14, 4.51**	**0.020**	0.69	0.37, 1.28	0.2

^1^ CI = confidence interval.

## Data Availability

Data presented in this manuscript are available from the corresponding authors on reasonable request.

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
