# Peer review of "Syphilis Infections, Reinfections and Serological Response in a Large Italian Sexually Transmitted Disease Centre: A Monocentric Retrospective Study"

_jcm, 2022, doi:10.3390/jcm11247499_

Round 1
Reviewer 1 Report
This is a good analysis of the data from the authors' hospital complex. It is a well-structured retrospective study that ticks all the boxes necessary in relation to methods and reporting of results.
I have four broad concerns with the article and a number of smaller points, some of which are related to the writing.
#1 English
The English has a number of errors that at times even confuse the meaning of sentences. Please get it reviewed by a native speaker. For example, in the second line of the Abstract, line 15, you say "far from being cleared", when I'm reasonably sure you meant "clear".
#2 . Nature of a Retrospective Study and Its Limits
You clearly state that "retrospective studies are subject to innate bias. (line 332 - 333). However, you do not elaborate on which types of bias have affected yours. This is quite important, especially with reference to the exclusions you show in Figure 1, which itself is well constructed. The discussion should include what those exclusions may mean for the conclusions. Of particular interest would be a comparison of those patients who did and did not have 12 month follow-up.
#3. Confounding Factors. Part of the problem with retrospective studies how to determine what are confounders that skew your results. You report in Table 3 that patients 40 years and above seemingly could skew the results, but in Table 5, age is not significant (p-values for non-response and serofast status above the alpha limit). So, where could age have an effect beyond the syphilis reinfections in Table 3 and how do you and other labs interpret that? Also, what are the other possible confounders the reader should be paying attention to?
#4. Discussion is Thin. You have many tables and statements of results but a very short discussion. I can't believe that is all you have been able to draw out of this data that readers would find important or interesting. This section needs to be beefed up.
The smaller points.
Table 1. I don't see the need for a p-value calculation here. You are simply describing the number of infections and reinfections per year. Because you have 9 levels (years), of course you will have a significant difference among the years. It's important statistically to know when we want to test hypotheses and when not. Here you are testing the hypothesis that there is a difference in the number of infections per year, but nowhere else do you indicate what the implication of that is.
Table 2. What are those dots in the margin of the Syphilis Stage section of the table? Probably they are extraneous, but if not what do they mean?
Lines 228 - 231. Here you do address the longer-term follow up of the patients. But, why does this result occur? The reporting is fine, but what does it mean?
Lines 305 - 306. You say 'this trend' and 'This last aspect', but in the previous sentence and paragraph you refer to a number of issues that could be that. Which do you mean?
Lines 340 - 341. You state that the sample size makes the study reliable. In this case, you should include in the results a power analysis. I would be particularly interested in what effect size, what ability your sample had to differentiate between the groups.
Lines 339 - 340. You state that no funding was received, but that you have an 'unconditional grant' from Gilead. A reader (like me) will see an inconsistency in these statements. What is Gilead providing in fact.
Reviewer 2 Report
Figure 1- There are 209 PWH who had one subsequent event of Syphilis infection. However, episodes of syphilis reinfection among them are 328. In addition, data is incomplete for 10 patients among 61 one subsequent event of Syphilis infection in HIV negative patients. However, the number become 54 for syphilis reinfection. Revise the figure.
Table 2- Though percentage of various stages of Syphilis during First Infection represented as 100% among HIV negative and HIV positive group, number of patients in each group is not accounted correctly.
Data in follow-up visits (months) in table 2 is confusing as study has two follow-ups (6- and 12-month duration). What is mean and median here?
Statement in line 204-207 looks contradicting. Based on the table 3 and Authors statement, homosexual/bisexual increased the risk of syphilis reinfections (line 204). However, Author’s also describes, sexual orientation was not associated to an increased risk of syphilis reinfection in PLWH (line 205-207).
Rewrite the sentence in line 212-216, “Among patients with syphilis reinfection effectively treated according to their syphilis stage at presentation and most recent guidelines, for which a scheduled follow-up visit at 6 and 12 months after treatment was available, no significant statistical difference was observed in serological response between PLWH and HIV-negative patients (p=0.2 at 6 months, p=0.1 at 12 months)” as it is difficult to understand.
I couldn’t understand the table 4. The way the numbers derived, and its percentage calculation is completely wrong. For example, 13 HIV negative patients showed serological response in 6 months and the percentage should be 24%, and the serological non-responders should be 41 (54-13).
Both non-trponemal and RPR terms were randomly used here and there in the manuscript. Follow either non-trponemal or RPR throughout the manuscript.
RPR or non-trponemal titre is used to monitor for a re-infection with syphilis and hence RPR tire cannot be used as a predictor for either serological response or non-response.
Title of the manuscript is Syphilis infections, reinfections, and serological response to treatment. However, the detail of treatment is found missing.
Round 2
Reviewer 2 Report
No comments